# Variability in Healthcare Expenditure According to the Stratification of Adjusted Morbidity Groups in the Canary Islands (Spain)

**DOI:** 10.3390/ijerph19074219

**Published:** 2022-04-01

**Authors:** Maria Consuelo Company-Sancho, Víctor M. González-Chordá, María Isabel Orts-Cortés

**Affiliations:** 1Health Promotion Service, Directorate General for Public Health, Canary Islands Health Service, 35003 Las Palmas de Gran Canaria, Spain; 2Nursing and Healthcare Research Unit (Investén-isciii), Institute of Health Carlos III, 28029 Madrid, Spain; 3Nursing Department, Universitat Jaume I, Avda Sos Baynat s/n, 12071 Castellon, Spain; vchorda@uji.es; 4Department of Nursing, University of Alicante (BALMIS), Alicante Institute for Health and Biomedical Research (ISABIAL), 03690 Alicante, Spain; isabel.orts@ua.es; 5CIBER of Frailty and Healthy Ageing, (CIBERFES) Institute of Health Carlos III, 28029 Madrid, Spain

**Keywords:** costs, variability, AMG, morbidity aggregator (or stratification systems), multi-morbidity

## Abstract

Morbidity is the main item in the distribution of expenditure on healthcare services. The Adjusted Morbidity Group (AMG) measures comorbidity and complexity and classifies the patient into mutually exclusive clinical categories. The aim of this study is to analyse the variability of healthcare expenditure on users with similar scores classified by the AMG. Observational analytical and retrospective study. Population: 1,691,075 subjects, from Canary Islands (Spain), aged over 15 years with data from health cards, clinical history, Basic Minimum Specialised Healthcare Data Set, AMG, hospital agreements information system and Electronic Prescriptions. A descriptive, bivariant (ANOVA coefficient η^2^) and multivariant analysis was conducted. There is a correlation between the costs and the weight of AMG (rho = 0.678) and the prescribed active ingredients (rho = 0.689), which is smaller with age and does not exist with the other variables. As for the influence of the AMG morbidity group on the total costs of the patient, the coefficient η^2^ (0.09) obtains a median effect in terms of the variability of expenditure, hence there is intra- and inter-group variability in the cost. In a first model created with all the variables and the cost, an explanatory power of 36.43% (R^2^ = 0.3643) was obtained; a second model that uses solely active ingredients, AMG weight, being female and a pensioner obtained an explanatory power of 36.4%. There is room for improvement in terms of predicting the expenditure.

## 1. Introduction

Appropriate management of chronic patients would improve health outcomes, boosting the sustainability of healthcare systems [1]. In Spain, the National Health System is financed with public funds through general taxation. The service is free, while prescribed pharmaceutical products entail co-payment by the patient. There are 17 autonomous regional communities, each with its own department of health that mainly controls the funding, organisation and provision of healthcare services in each territory, of which the Canary Islands are one.

The World Health Organization (WHO) defines comorbidity as the presence of two or more diseases that occur in a concomitant manner with a primary disease [2], with the possibility that the progress of both may worsen. Its prevalence increases progressively with age, between 60% and 70% among people over the age of 60 and 80 years, respectively [3]. It is a determining factor in accounting for the consumption of healthcare resources in chronicity [4], insofar as the more comorbidity there is, the greater the consumption of resources [5]. A tool that facilitates the analysis of this comorbidity and multi-morbidity will allow a better comprehension of the use, costs, and quality of the combination of services received in a population [4]. The strategies developed internationally and nationally to cope with chronicity propose risk stratification, which will help clinicians in their decision-making [6,7]

Stratifiers, also known as risk adjustment models, are a system for measuring morbidity, which classifies people into mutually exclusive clinical categories, based on information obtained from clinical histories. Taking comorbidity as the starting point, stratifiers introduce the concept of levels of complexity, attempting to predict the risk of occurrence of a given event or a certain level of resource consumption. Stratifiers use inputs, such as medical diagnoses, age, sex, contact with the healthcare system, surgical procedures or pharmaceutical prescription, to obtain scores that make it possible to classify the population into homogeneous morbidity groups. Usually, they do not take into account factors such as personal circumstances, social context, fragility, resilience to problems or nursing care, although care is regarded as a key element in strategies for dealing with chronicity [6].

There are different developments and their purpose is multifaceted, from identifying high-cost patients for case management, monitoring health status, determining patterns of resource use and consumption, or predicting population risk and cost, which is useful for pharmaceutical budget allocation. [8,9]. In short, they are all management tools.

In Spain, over the last 15 years, prolific research has been conducted on the functioning of ACGs and GRCs in some autonomous communities for various purposes, such as cost measurement [10,11], cost prediction [12], effectiveness of Primary Care (PC) referrals [13], such as morbidity stratification [14], analysis of excessively frequent use of services [15] and pharmaceutical dispensing [11,12,16,17]. This may be due to the rise in the phenomenon of chronicity, where healthcare systems have prioritised the detection of patients with greater complexity, in order to better manage services and resources. Adjusted Morbidity Groups (AMGs) are a risk adjustment system adapted to the Spanish healthcare environment that allows the general population to be stratified into morbidity groups. It has been constructed from qualitative–quantitative models where the care needs of users are collected according to variables such as mortality, risk of admission, Primary Care visits, or prescription. Its results can be used both in clinical-care management, as well as in epidemiology and healthcare administration. This system is based on the presence of chronic diseases, integrating all available medical diagnoses from the Primary Care medical history, and also considers recent acute diagnoses, through the Basic Minimum Hospital Data Set (BMHDS), as well as the date of diagnosis. It classifies the population into six morbidity groups, divided in turn into five levels of complexity, plus a healthy population group. Thus, the population is grouped into 31 mutually exclusive categories [18]. In addition, it provides additional results at an individual level, including the number of chronic diseases, the number of organ systems affected by a chronic disease, a clinical summary label, and the AMG weight of complexity or multi-morbidity index. The multi-morbidity index is calculated as the sum of the weights assigned to each of the pathology groups that affect the patient (groups that have an individual weight ranging from 0.14 for attention deficit disorder to 4.99 in the case of multiple myeloma). It is the population aggregator chosen by the Ministry of Health, Consumption and Social Welfare for the implementation of the Strategy for the Approach to Chronicity in the Spanish National Health System. The AMG system has been used in 13 Autonomous Communities, with a positive assessment of the results obtained [19]. 

The AMG system performs highly in accounting for relevant health outcomes and can be useful for clinical practice, resource planning and public health research [20]. Some of the studies carried out with this aggregator prove its concordance between the classification and the level of allocation by family doctors [21,22]; its capacity as a stratification tool in different chronic pathologies [23,24,25] and its value to refine the classification of more complex patients [26,27]. Furthermore, it has been compared with other stratifiers [27,28,29] proving to be, in general, a better predictor than these in relation to the use of certain healthcare resources (PC and Specialised Care (SC) consultations, emergencies, admissions). However, we found no studies that relate it to total costs.

Cost and stratifier studies such as the AMG focus their objectives on explaining cost variations, predicting consumption, building predictive models for budget allocation, obtaining the relative weight of costs, improving budget allocations, or analysing predictive capacity by adding other variables. They assume that the cost classifications in the stratification are adjusted to the expenditure per patient. In this study, we wanted to find out whether healthcare expenditure is similar for users with similar scores in the Adjusted Morbidity Groups classification system, or if on the contrary there is variability not explained by the AMG calculations. The objective was to analyse the variability of healthcare expenditure, within each of the groups generated by the stratifier and among different groups.

## 2. Materials and Methods

### 2.1. Design and Scope

An analytical and retrospective observational study based on data from health information system records. The study was carried out in the autonomous community of the Canary Islands (Spain) taking data from the years 2017 and 2018. Predictor variables were considered as of 31 December 2017, and total expenditure was calculated in 2018.

### 2.2. Participants

The study population comprised people over 15 years of age assigned to the health card database of the Canary Islands Health Service (SCS) throughout the year 2018 (1,691,075). The exclusion criteria were being under 15 years of age or being a mutual insurance society member (patients cared for by State mutual societies that provide social benefits and healthcare solely to their officials).

Of the study population, 51.04% (*n* = 863,071) were female and the mean age was 46.6 years (SD 17.9). A total of 17.39% (*n* = 294,032) of the population were over 65 years of age, and 23.21% (*n* = 392,520) were pensioners. The most frequent level of pharmaceutical contribution was 40%, with a total of 55.28% (*n* = 934,798).

In the analysis by sex it can be seen that 19% (*n* = 164,316) of females were over 65 years of age compared to 15.7% (*n* = 129,676) of males. Regarding the type of social security membership, 25.1% (*n* = 216,630) of females were pensioners compared to 21.2% (*n* = 175,536) of males. Regarding the percentage of pharmaceutical contribution, females have higher percentages than males in the 0% groups (*n* = 88.644 in females vs. *n* = 56,444 in males) and 10% (*n* = 187,175 vs. *n* = 160,439), and lower percentages in the rest.

### 2.3. Variables and Information Sources

The outcome variable was cost, calculated for the whole of 2018, of consultations and emergencies in public and state-subsidised private centres, hospitalisations, major outpatient surgery and prescriptions dispensed in pharmacies (Appendix A).

The sociodemographics, obtained on 31 December 2017, included sex, age, type of social security membership, percentage of pharmaceutical contribution; predictor variables related to the AMG adjustment system (complexity weight and AMG morbidity group); and clinical variables (Barthel index < 60, Pfeiffer questionnaire ≥ 5, having 2 or more hospital admissions in the last 12 months, and the number of active pharmaceutical ingredients) (Appendix A).

Different data sources were used to obtain the information depending on its nature. The variables sex, age, type of Social Security membership and type of pharmaceutical contribution were obtained from the health card database of the Canary Islands Health Service. The variables of medical consultations in the Primary Care centre, medical consultations at home, nursing consultations in the Primary Care centre, nursing consultations at home, and outpatient emergencies in public centres were obtained from the Primary Care Health History. The variables of hospital emergencies in public centres and specialised care medical consultations in public centres were collected from the Clinical History of Hospital Care. The variables of hospitalisations in public centres, home hospitalisations in public centres, episodes of outpatient major surgery in public centres, hospitalisations in state-subsidised private centres, episodes of major outpatient surgery in state-subsidised private centres were obtained from the Minimum Basic Data Set of the Specialised Care Register (CMBD-RAE). The variables of state-subsidised private emergencies, specialised care medical consultations in state-subsidised private centres, physiotherapy treatments in state-subsidised private centres and rehabilitation consultations in state-subsidised private centres were collected from the State-Assisted Hospitals Information System (SICH). Finally, the data on prescriptions dispensed in pharmacies charged to the Canary Islands Health Service, whether electronic or on paper, were obtained from the Electronic Prescription of the Canary Islands Health Service (REC-SCS).

The key unit of integration of all the variables from the different health information systems was the patient’s file number, which was encrypted and transformed into a single code for each individual.

### 2.4. Statistical Analysis

The statistical analysis was performed using the statistical program R Commander 2.6-2, graphical user interface of the statistical package R 4.0.0. The level of significance of *p* < 0.05 was adopted. An analysis consisting of descriptive statistics and bivariate analysis to obtain the relationship between explanatory and outcome variables was carried out, including an analysis of variance to study the effect of morbidity groups on costs, and a multiple linear regression to study the relationship between the weight of complexity and cost.

First, a descriptive analysis of all the global and group study variables, as well as by sex, was carried out to describe the characteristics of the study subjects. Secondly, a bivariate analysis was performed to study the relationship between each independent variable and the use of resources in the year immediately after. We used corresponding evidence to compare quantitative variables in two or more groups. In the case of the bivariate association study between qualitative variables, the Ji-square test or Fisher’s exact test was used. For the association between quantitative variables, the Student-Fisher t-test (two groups) or ANOVA (more than two groups) were used. In the event that the conditions for the application of the parametric tests were not met, the corresponding non-parametric tests were used.

Specifically, to calculate the influence of the AMG morbidity group on the patient’s total costs, we then conducted an analysis of the variability between groups and intra-groups by means of an analysis of variance (ANOVA), using the coefficient η^2^ [30] as follows:The sum of the factor squares (between groups) measures the variability between factor levels.The sum of the residual squares (intra-group) measures the variability within each level, that is, the variability that is not due to the qualitative variable or factor. It is calculated as the sum of the squares of the deviations of each observation with respect to the average of the level to which it belongs.The sum of total squares measures the total variability of the data and is defined as the sum of the squares of the differences of each observation with respect to the general average of all observations.To define the magnitude of the effect with this coefficient η^2^, we used a convention in which >0.01: small, >0.06: medium, >0.14: large.

Finally, to study the association between the explanatory variables and the outcome variables, a multiple linear regression model was constructed taking total costs as a dependent variable. The logarithm was applied to normalise the cost since it does not follow a normal distribution.

### 2.5. Ethical Considerations 

The ethical principles of the Declaration of Helsinki and the standards of Good Clinical Practice in research were followed. The study was approved on 29 January 2021, by the clinical research ethics committee of the Dr. Negrín University Hospital of Gran Canaria (Code: 2021-037-1). The study complies with the provisions of Organic Law 3/2018, of 5 December, on the Protection of Personal Data and guarantee of digital rights, as well as Regulation (EU) 2016/679 of the European Parliament and of the Council of 27 April 2016, on Data Protection.

## 3. Results

### 3.1. Descriptive Analysis

#### 3.1.1. Clinical Variables

The distribution of the study population by morbidity, sex and age is shown in Table 1. As age increases, comorbidity increases: the increase in females compared to males in the group affected by four or more systems is notable. The complexity of our population, according to the AMG, has an average weight of 5.82 (SD 5.89) with quartiles of 1.63, 4.13 and 8.14.

A total of 1.35% (*n* = 22,800) of the population had two or more hospital discharges in 2017 and 0.37% (*n* = 6308) of the population had a Barthel index of less than 60 points (with 1.36% (*n* = 22,998) of the population assessed at least once), and 0.43% (*n* = 7276) had a Pfeiffer index greater than or equal to 5 points (with 3.37% (*n* = 56,989) of the population assessed). The mean number of drugs prescribed in the population was 4.35 (SD 5.1) and 36.92% (*n* = 624,344) were polymedicated patients (with 5 or more drugs).

Broken down by sexes, the mean (6.52) weight of AMG complexity was higher in females compared with males (5.10). A total of 0.5% (*n* = 4315) of females had a Barthel index of less than 60 points compared to 0.2% (*n* = 1656) of males. In the Pfeiffer questionnaire greater than or equal to five points, 0.6% (*n* = 5178) of females were observed compared to 0.2% (*n* = 1656) of males. A total of 1.4% (*n* = 12,082) of females had two or more incomes in 2017 compared to 1.3% (*n* = 10,764) of males. Females consumed an average of 5.19 active pharmaceutical ingredients compared to 3.48 for males. Patients with two or more hospital discharges were older than those who did not meet this criterion (61.5 vs. 46.4 years), the same occurred with those who had a Barthel index under 60 versus those who did not (79.9 vs. 46.5 years) and those who had a Pfeiffer index greater than or equal to 5 (80.0 vs. 46.5 years). In addition, age was positively correlated with AMG weight (Spearman’s rho = 0.47). Table 1 shows the mean ages of each morbidity group.

#### 3.1.2. Average Costs

The cost analysis produced an average total cost per patient in 2018 of €1283.08 (SD 3007), with the highest item being Specialised Care (€571.90, SD 2274) followed by the dispensing of prescription drugs (€345.52, SD 946), Primary Care (€246.85, SD 421) and state-subsidised private centres (€118.80, SD 808). The average total cost for females was €1377.04 versus €1185.14 for males.

### 3.2. Bivariate Analysis

The bivariate analysis between the total cost of the year 2018 and the quantitative variables showed a significant correlation with the AMG weight (rho = 0.678) and the prescribed active ingredients (rho = 0.689), but not with age (rho = 0.386).

In relation to the categorical variables, the average total cost for females was €1377.04 (SD 2868.27; CI 95%: 1370.99–1383.09) vs. €1185.14 (SD 3142.05; CI 95%: 1178.37–1191.91) in males; €2950 (SD 2469.06; CI 95% 2934.28–2966.34) in those over 65 vs. €932 (SD 4435.34; CI 95% 928.09–936.28) in those under that age; €2880 (SD 4575.85; CI 95% 2865.98–2894.61) in pensioners vs. 800 (SD 2107.49; CI 95% 796.66–803.91) in active subjects; the differences are even greater between people with a Barthel index under 60 compared with the rest (€6058 (SD 6079.87; CI 95% 5908.14–6208.27) vs. €1265 (SD 2975.15; CI 95% 1260.71–1269.69)), people with a Pfeiffer index greater than or equal to 5 compared to the rest (€4877 (SD 4975.91; CI 95% 4762.35–4991.05) vs. €1268 (SD 2986.27; CI 95% 1263.04–1272.06)) and among people who have had two or more discharges in the previous year than those who have not (€5808 (SD 2818.28; CI 95% 5700.65–5915.84) vs. €1221 (SD 8288.82; CI 95% 1216.96–1225.51). Where there does not seem to be a linear relationship is between the percentage of pharmaceutical contribution and total costs (Figure 1), noting that a lower contribution reflects higher costs.

Regarding the relationship between morbidity groups and costs (Table 2), the ANOVA analysis found significant differences between the total mean expenditure of the different morbidity groups of the AMG system, although each of the morbidity groups had a high variability internally (coefficient of variation greater than 1). Otherwise, the result of the coefficient η^2^ in the ANOVA analysis between groups through the sum of the squares of the factor (1,393,131,734,896), divided by the sum of the total squares: (1,393,131,734,896 + 13,897,147,949,253 = 15,290,279,684,149) stood at 0.09, and the influence of the Morbidity group AMG on the total costs of the patient, therefore, had a magnitude of medium effect (Table 3).

### 3.3. Multivariant Analysis

In a first linear regression, the explanatory variables include: high in 2017 greater than or equal to two, Barthel index under 60, age, being female, being a pensioner, AMG weight, Pfeiffer equal to or greater than 5, percentage of pharmaceutical contribution and number of active ingredients prescribed were significantly related to the logarithm of the total cost, accounting for 36.43% (R^2^ = 0.3643; *p* < 0.001) of the model (Table 4). The AMG alone in relation to cost has an R^2^ = 0.31.

In a second model constructed with the number of active ingredients, AMG weight, being female and being a pensioner, which were the variables that most correlated with costs in the previous model, the variability of the total cost algorithm was explained by 36.4% (*p* < 0.001) (Table 5).

## 4. Discussion

The study population was younger than the Spanish average, with 19.58% of people over 65 years of age [31], with a low income since more than 50% had a level of pharmaceutical contribution of 40%, which is equivalent to an income of less than €18,000 per annum, and with higher morbidity than that described in the validation study of the AMG [18]. Furthermore, the average number of drugs prescribed in our population was 4.35, above that found by a similar study of 3.40 [32].

On the other hand, in the descriptive analysis it was observed that the female population presented more functional and cognitive impairment, a greater degree of complexity, consumed more active ingredients and had more hospital admissions than males, which is consistent with other studies [7]. These results and the underdiagnosis of disability [33], are aspects that would require a specific analysis.

However, there was a correlation between the total cost and the weight variables of AMG and the morbidity group, a finding that coincides with other studies where the average total healthcare expenditure increased with the number and intensity of chronic diseases [34]. This correlation also occurred with prescribed active ingredients and to a lesser extent with age, as seen in other studies [35], and was not significant with the rest of the variables. Another variable generated by the AMG is the weight of complexity which, in relation to costs, is only the second most important factor, the first being the number of prescribed active ingredients, which comes after other variables such as being female and a pensioner.

In this study, it was observed that the average cost assigned to each of the seven groups of morbidity is different, increasing with complexity, with the exception of the pregnancy and childbirth group. However, within each group, there was a certain degree of heterogeneity since the coefficient of variation was higher than one in all cases. All this was reflected in an η^2^ value that indicated a medium effect of this variable on the variance of the total cost, and we could therefore deduce that the morbidity group assigned to each patient was not sufficient to explain their costs. These results were difficult to compare since we could not review other studies that valued this metric of relationship with the total costs for any stratifier.

An adjustment system can be considered acceptable if it can explain between 20% and 25% of the variance between individuals in resource use [36]. In our case, the AMG system did not by itself achieve this explanatory power, but both the model created in our study and the reviewed studies shown below exceeded this value. Even so, it would be interesting to approximate it to the maximum of adjustment by adding other inputs given the scarcity of resources and the importance of equity for a healthier society. More complex (more comprehensive) models of risk adjustment would better predict cost and would have better performance [37].

The explanatory power of the four variables of our model on costs (female, pensioner, AMG weight and prescribed active ingredients) was 36.4%. Comparison with other studies is difficult because they use different stratifiers that measure morbidity and severity, but not with the same inputs or algorithms, and use different types of research and analysis. Although according to several authors, these stratifiers present similarities, we believe that they are only approximately comparable, but for the purpose of our research, we find it interesting to comment on them. For example, in a study where the CRG was used including a population of all ages, the model composed of state of health, severity, age and sex reached an R^2^ of 49.4% of total expenditure explanation [34]. Another study, which also analysed a sample of all age groups, found a predictive model consisting of age, sex, aggregate diagnostic groups (ADGs) and Rx-MGs (drugs) obtained an R^2^ of 38.2%, which increased to 46.5% when adding the total cost of the previous year [38]. In another study, conducted only in a primary care setting, the model that included age, sex and ACG obtained an explanatory power R^2^ of 42.6% (*p* < 0.001) of the total expenditure while the ACG classification explained the variance of this total cost of primary care by 51.6% [10]. In the Orueta study, where the analysis was carried out with the entire population aged over 14 years of the community, using two-stage models and hierarchical models, three stratifiers–ACG-PM, CRG and DCG-HCC–were compared, obtaining the best R^2^ with respect to the total costs, a model that adds the previous costs to the stratifier data. The results were 26.9% for DCG-HCC, 26% for ACG and 25.4% for CRG (which in this case did not include prescription information) [32]. Coderch’s work constructed a linear regression model with age, ACRG3 (part of the CRG classification), previous expenditure in pharmacies and expenditure on hospital medication, achieving an R^2^ of 38.2% [35].

The strengths of our study include the large sample size, the inclusion of the cost of primary care, and both public and private hospitals with state funding. Another strength is that we analysed data from more recent years than in the studies reviewed, which meant an improvement in the number of records and the maturity of the information systems. However, it is necessary to clarify some limitations in our study, such as the lack of a standardised method to calculate the cost per patient in our healthcare system, which meant we had to calculate them based on preset prices, accepting that there are some services that are not clearly identified as home hospitalisation, mental health or day hospital, inter alia. In addition, this study does not use the previous total costs, which, according to Chang [39], are a more powerful tool than stratification in terms of correctly identifying users with high expenses, stating that there is still much room for improvement in diagnostic-based models. Another aspect to consider is the low number of functional assessments (Barthel index) and cognitive assessments (Pfeiffer test) carried out since they are two indicators of quality of life that can provide a more accurate picture of the state of health of our population with multi-morbidity.

Another general limitation of stratifiers that needs to be considered is that the quality of the results is subject to the quality of the clinical history records. In a study with ACG to estimate costs in PC, the weighting of the ACG seems to be sensitive to the precision with which doctors include diagnoses in clinical histories [40]. This may mean that various populations and studies are not comparable. Depending on the sensitivity of health policies, pathologies are underestimated or overestimated by professionals who are more sensitised to the health programmes offered by their Health Services. It is also known that in primary care there is usually an under-reporting of diagnoses, although this disadvantage does not appear to be important enough to affect the results since a requirement of these adjustment models is to work with imperfect data in a real world [32].

The studies reviewed show that the inputs added to the stratifiers are insufficient to predict the cost beyond 50%, and often adding other factors to the stratifier such as certain clinical variables and previous costs [35,38]. Although the information on morbidity and co-morbidity is included, the heterogeneity of clinical cases makes future cost estimates difficult [36]. In fact, all these stratifiers stem only from medical diagnoses, age, sex, contacts with the healthcare system, surgical procedures or pharmaceutical prescription, and do not take account of determinants for health such as personal circumstances including education, purchasing power, lifestyles, social context, fragility, resilience to problems or nursing care. All of these factors may be causes of unexplained variability. If the central role of the aggregators is to define what complexity is, we need to add other inputs so that the algorithm refines its classification. Future research should be along these lines to improve these tools that help managers and healthcare professionals. The creators of the AMG themselves comment on the need to include other factors such as psychosocial problems, which would add greater value to the multifactorial equation [18]. As Starfield [41] states, the burden of morbidity interacts with social and economic factors, which presents a challenge to measure this complexity.

The AMG is a very recent stratification system, and future research will be able to approximate data. Specifically, our study proves that the variability of healthcare expenditure in users with similar AMG scores is significant. Future studies may improve the accuracy in classifying people through the inclusion of inputs that are not currently considered and that also determine the complexity, such as lifestyle habits, risk factors, complying with healthcare recommendations, socioeconomic conditions, place of residence, unwanted solitude, or nursing care. We know that nursing care has been shown to influence the prediction of total expenditure by improving the explanatory power of resource use from 13.70% (sociodemographic variables) to 19.77% when nursing diagnoses are factored in [42].

## 5. Conclusions

There is variability in expenditure among the seven groups classified by the AMG, which is consistent when classified by morbidity. However, there is also high variability in users with similar scores and classified in the same AMG group, which may indicate that the stratifier needs to be improved. The model created in the study explains 36.4% of the variability in total expenditure, matching the predictive ability of the adjustment models, which remains small to medium. Morbidity will be the marker of expenditure in public and private health care systems, so an improvement of the stratifiers should be taken into account.

There have already been some attempts such as the inclusion of socio-demographic variables or functional and cognitive assessment, readmissions and prescribed therapeutic groups. This room for improvement may contribute to the more equitable distribution of resources, greater satisfaction of professionals when they see that their work is more accurate, and a more precise tool for managers.

## Figures and Tables

**Figure 1 ijerph-19-04219-f001:**
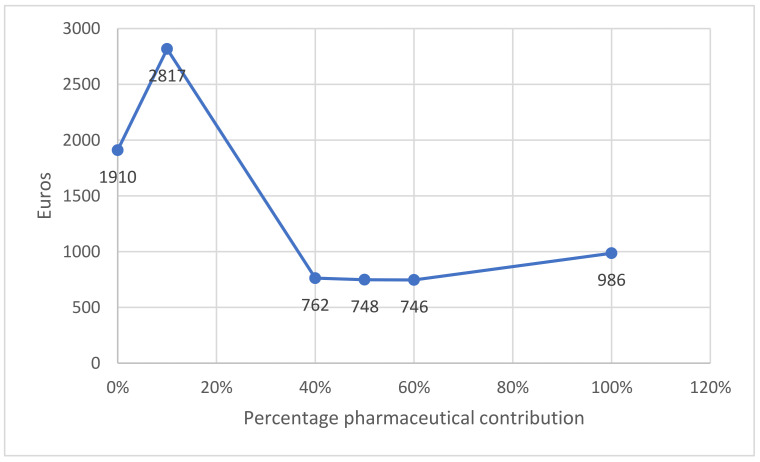
Relationship between pharmaceutical contribution and total costs in euros.

**Table 1 ijerph-19-04219-t001:** Distribution of the population by morbidity group, sex and average age.

	Total Population	Males	Females	Average Age (SD)
Morbidity Group	% (*n*)			
Healthy population	10.28 (173,846)	13.2 (22,948)	7.5 (13,038)	41.3 (14.9)
Acute illness	6.43 (108,798)	8.3 (9030)	4.7 (5113)	36.0 (12.3)
Pregnancy or childbirth	1.81 (30,591)	0.0	3.5 (1071)	32.6 (6.6)
Chronic illness in 1 system	17.18 (290,462)	20.5 (59,545)	14.0 (40,665)	39.4 (14.0)
Chronic illness in 2 or 3 systems	32.12 (543,237)	33.1 (179,811)	31.2 (169,490)	44.5 (16.6)
Chronic illness in 4+ systems	30.70 (519,212)	23.5 (122,015)	37.6 (195,224)	56.9 (18.1)
Active neoplasia	1.47 (24,929)	1.4 (349)	1.5 (374)	64.1 (14.5)

**Table 2 ijerph-19-04219-t002:** Average costs by morbidity group in euros.

Morbidity Group	Average Cost in Euros	Standard Deviation	Coefficient of Variation
Healthy population	169	1.087	6.45
Acute illness	425	1.482	3.48
Pregnancy or childbirth	1718	2.527	1.47
Chronic illness in 1 system	517	1.632	3.16
Chronic illness in 2 or 3 systems	981	2.436	2.48
Chronic illness in 4+ systems	2419	4.002	1.65
Active neoplasia	4126	6.148	1.49

**Table 3 ijerph-19-04219-t003:** Calculation of η^2^ values in the analysis of variance (ANOVA).

Df	Sum Sq	Mean Sq	F Value	Pr (>F)
GM_C *	6	1,393,131,734,896	232,188,622,483	<2 × 10^−16^ ***
Residuals	1,691,068	13,897,147,949,253	8,217,971	

* GM_C, Variable Adjusted Morbidity Groups; *** Raised to 16.

**Table 4 ijerph-19-04219-t004:** Relation between total cost and independent variables.

	Coefficients	Standard Error	T Value	*p*
Constant	3.485	0.007	439.162	<0.001
More than 2 discharges in 2017	−0.161	0.015	−10.754	<0.001
Barthel index less than 60 points	−0.424	0.029	−14.488	<0.001
Age	−0.002	<0.001	−21.125	<0.001
Female	0.253	0.003	73.254	<0.001
Pensioner	0.276	0.006	41.026	<0.001
AMG weight *	0.133	0.0004	281.994	<0.001
Pfeiffer index over 5 points	−0.039	0.027	−1.447	<0.001
Percentage of pharmaceutical contribution	0.0005	<0.001	3.585	<0.001
Amount of prescribed active ingredients	0.183	<0.001	341.926	<0.001

* AMG: Adjusted Morbidity Groups; Residual standard error: 2.211 on 1691066 degrees of freedom Multiple R-squared: 0.3643, Adjusted R-squared: 0.3643. *p*-value: <0.001.

**Table 5 ijerph-19-04219-t005:** Relationship between costs and most significant explanatory variables.

Coefficients	Estimate	Std. Error	T Value	Pr (>|t|)
(Intercept)	3.4101391	0.0028512	1196.02	<0.001
Female	0.2553705	0.0034543	73.93	<0.001
Pensioner	0.2073515	0.0047793	43.38	<0.001
Weight of the AMG *	0.1310863	0.0004630	283.15	<0.001
Active ingredients prescribed	0.1818807	0.0005296	343.44	<0.001

* AMG: Adjusted Morbidity Groups; Residual standard error: 2.212 on 1691071 degrees of freedom. Multiple R-squared: 0.364, Adjusted R-squared: 0.364. *p*-value: <0.001.

## Data Availability

Not applicable.

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
