# Peer review of "Variability in Healthcare Expenditure According to the Stratification of Adjusted Morbidity Groups in the Canary Islands (Spain)"

_ijerph, 2022, doi:10.3390/ijerph19074219_

Round 1
Reviewer 1 Report
Dear Authors,
Please find attached the review report.
Best regards,

Reviewer 2 Report
Dear author,
- why do the authors present the years 2017 and 2018 in the study? please justify your choice,
- Aren't two years a short period to compare and draw conclusions?
- I recommend finalizing the conclusion of the article
- references not processed according to journal instructions, edit.
Reviewer 3 Report
The paper is focused on healthcare costs model, developing an interesting
integrated approach.
The approach is well explained and the example allows readers to understand the whole problem, and also to replicate it.
Despite the paper is well presented and structured, I suggest introducing the main results in the abstract in brief, so that it is possible to have an idea about the overall ones.
Moreover, the introduction, in the presented form, is quite negligible. I suggest restructuring it explaining in-depth the problem.
The English form has to be revised for the presence of some typos. The literature review appears not completely updated.
